# A Novel Meander Bowtie-Shaped Antenna with Multi-Resonant and Rejection Bands for Modern 5G Communications

Yanal S. Faouri [1], Sarosh Ahmad [2,3,*], Naser Ojaroudi Parchin [4], Chan Hwang See [4] and Raed Abd-Alhameed [5]

1   Department of Electrical Engineering, The University of Jordan, Amman 11942, Jordan; y.faouri@ju.edu.jo
2   Department of Signal Theory and Communication, Universidad Carlos III de Madrid, Leganes, 28911 Madrid, Spain
3   Department of Electrical Engineering and Technology, Government College University Faisalabad, Faisalabad 38000, Pakistan
4   School of Engineering and the Built Environment, Edinburgh Napier University, Edinburgh EH10 5DT, UK; n.ojaroudiparchin@napier.ac.uk (N.O.P.); c.see@napier.ac.uk (C.H.S.)
5   Faculty of Engineering and Informatics, University of Bradford, Bradford BD7 1DP, UK; r.a.a.abd@bradford.ac.uk
*   Correspondence: saroshahmad@ieee.org

**Abstract:** To support various fifth generation (5G) wireless applications, a small, printed bowtie-shaped microstrip antenna with meandered arms is reported in this article. Because it spans the broad legal range, the developed antenna can serve or reject a variety of applications such as wireless fidelity (Wi-Fi), sub-6 GHz, and ultra-wideband (UWB) 5G communications due to its multiband characterization and optimized rejection bands. The antenna is built on an FR-4 substrate and powered via a 50-$\Omega$ microstrip feed line linked to the right bowtie's side. The bowtie's left side is coupled via a shorting pin to a partial ground at the antenna's back side. A gradually increasing meandering microstrip line is connected to both sides of the bowtie to enhance the rejection and operating bands. The designed antenna has seven operating frequency bands of (2.43–3.03) GHz, (3.71–4.23) GHz, (4.76–5.38) GHz, (5.83–6.54) GHz, (6.85–7.44) GHz, (7.56–8.01) GHz, and (9.27–13.88) GHz. The simulated scattering parameter $S_{11}$ reveals six rejection bands with percentage bandwidths of 33.87%, 15.73%, 11.71 7.63%, 6.99%, and 12.22%, respectively. The maximum gain of the proposed antenna is 4.46 dB. The suggested antenna has been built, and the simulation and measurement results are very similar. The reported antenna is expanded to a four-element design to investigate its MIMO characteristics.

**Keywords:** multi-band; UWB; 5G communications; sub-6 GHz; notches; bowtie-shaped; multiband; MIMO; time-domain analysis

## 1. Introduction

Modern wireless communication devices, which have evolved fast over the last four decades, are required to support a variety of applications, including real-time voice communication, text messaging, Wi-Fi, Bluetooth, Global Positioning System (GPS), video apps, and among others. All these applications operate in distinct frequency bands, necessitating the use of frequency reconfigurable antennas or multiband antennas to handle several applications with a single antenna. Frequency diversity can be configured electronically by utilizing varactors [1–3], micro-electromechanical systems (MEMS) [4], PIN diodes [5,6], or liquid metal [7]. On the other hand, designing an antenna in which its reflection coefficient spans the UWB range has grown in popularity, for its lowliness, inexpensive production costs, small power consumption, simplicity of production, and large bandwidth; so, the federal communication commissions (FCC) allowed UWB to use the unlicensed operating band from 3.1 to 10.6 GHz in 2002 [8]. Several UWB antennas utilize this band completely,

as in [9], or obtained the other definition of the UWB by covering more than 500 MHz of bandwidth, as in [10].

Numerous distinct geometries of multiband and or UWB antennas with one or more notches have been documented in the literature. A lanky leaf antenna was reported in [11] to produce two rejection bands within the UWB range of 2.8–10 GHz. A small monopole antenna was adjusted in [12] to produce triple rejection bands within the UWB of 2.285–19.35 GHz with 5.88 dB gain and utilizes three PIN diodes. Four nested hexagonal fractal antennas were designed in [13] to operate in multiple frequency bands. For one of their designs, 'Antenna III' has five resonant frequencies in the UWB of 1.92–13.45 GHz with only one notch at a level around −6 dB and a peak average gain of 2.96 dB. A hexagonal shape microstrip patch with its edges being replaced by round curves operating in the frequency range 3–27.57 GHz [14] was adjusted by adding an inverting stub and other two slots of triangular shape to create triple frequency notches with bandwidths of 1.63 GHz, 1.09 GHz, and 0.76 GHz in an SWB with a 179.4% percentage bandwidth (PBW) [15]. In [16], a monopole antenna based on the split-ring resonator technique is well-designed to produce three operating bands. The antenna has a resonance at 2.45 GHz in addition to dual bands within the UWB range 3.4–11.8 GHz. A unit-cell metamaterial of dimensions $10 \times 10$ mm$^2$ was extended into a $2 \times 1$ and $2 \times 2$ MIMO in [17]. These models provide two distinct bands to operate in the s- and x-band with resonances at 4.27 GHz, 5.42 GHz, and 12.4 GHz. A single-sided bowtie or a monopole bowtie in one layer with meander arms on the other layer was investigated in [18] to operate at three resonating bands. An isosceles triangular microstrip antenna coupled electromagnetically to unequal an arms V-shaped parasitic has been proposed in [19] to provide triple spectrums. This configuration can support six resonating bands at resonant frequencies of 2.88 GHz, 3.64 GHz, 3.95 GHz, 4.38 GHz, 4.81 GHz, and 5.6 GHz with five rejection bands between them where two of these rejection bands have a reflection coefficient level larger than 5 dB. Four spectrums for sub-6 GHz and mm-wave applications that have been radiated from a slotted patch of conical shape connected to a small triangular patch were proposed in [20]; the design resonant frequencies were 2.4 GHz, 5.2 GHz, 5.8 GHz, and 27.5 GHz. An antenna element of the 'F' shape placed above a truncated ground plane has been proposed in [21] to operate in four reconfigurable frequency modes.

The design and exploration of a UWB/multi-band 5G antenna with Hexa-frequency stopping bands and seven passing bands are presented in this work, incorporating a meander line structure that can be also integrated into a designing coupler [22] or power divider [23]. This design incorporates both multiple resonant modes and frequency rejection strategies to provide seven operational bands. The antenna was designed in the style of a bowtie, with each side made up of a meander line. The antenna exhibits a directional pattern with a major lobe and back lobe at lower frequencies as an alternative to the normal omnidirectional radiation pattern that would be formed from a similar limited ground arrangement. The proposed antenna has been further investigated compared to [24] by addressing an equivalent circuit model, and the circuit components' value effect is parametrized in addition to the time-domain analysis of the output signal. The suggested antenna has been manufactured, and the similarity between simulation and measured results were observed. Finally, an expansion of the designed antenna has been extended to form a four-port MIMO antenna to investigate the mutual coupling along with other MIMO characteristics at all resonating bands.

## 2. Proposed Antenna Design

A full parametric study is conducted on all variables to acquire the dimensions reported in Table 1. The proposed antenna with allocated parameters is illustrated in Figure 1. The antenna was built on a double-sided FR-4 substrate with a dielectric constant of 4.4 and a loss tangent of 0.02. The antenna's overall dimensions are $30 \times 30 \times 1.6$ mm$^3$. The right side of the bowtie is linked to the feed line, while the other portion of the bow tie is grounded using a shorting via to enhance the resonances since without the shorting

pin; only two resonances are developed. Both bowtie-shaped resonators are located at the top layer as shown in Figure 1a where each side of the bowtie has a gradually increasing meander line arms. The arm's length, width, and spacing have been optimized to produce several resonances with an acceptable bandwidth for the utilized services and several notches with an acceptable rejection level to filter out other services. Figure 1b shows the modified ground plane that has been adjusted to sweep over the UWB scope. Figure 1c displays a side view of the board to indicate the shorting pin placement.

**Table 1.** Suggested Antenna Lengths in mm.

| Parameter | Value | Parameter | Value |
|-----------|-------|-----------|-------|
| WS | 30 | LS | 30 |
| LG | 11 | W1 | 1.5 |
| W2 | 2.5 | W3 | 2.8 |
| W4 | 0.6 | W5 | 0.4 |
| W6 | 0.5 | W7 | 1 |
| W8 | 13.2 | W9 | 15.3 |
| L1 | 18.75 | L2 | 13.25 |
| L3 | 9.25 | L4 | 1.75 |
| L5 | 4.25 | L6 | 6.75 |
| L7 | 0.25 | L8 | 1.25 |
| L9 | 5.25 | L10 | 23.25 |

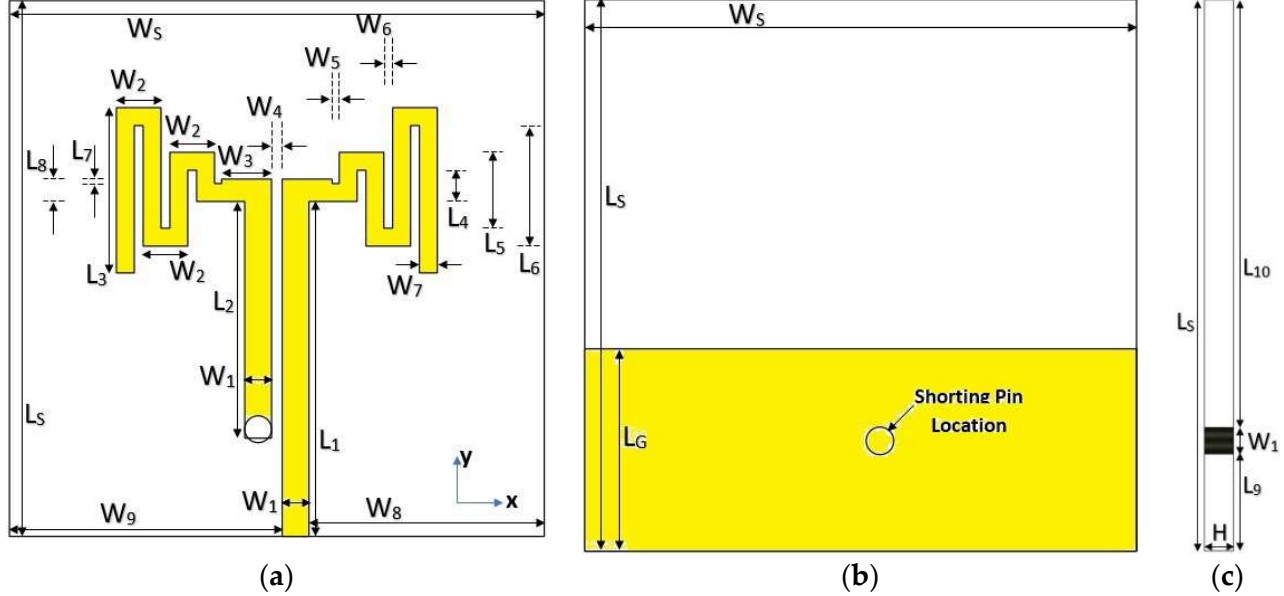

**Figure 1.** The suggested antenna; (**a**) front view, (**b**) back view, and (**c**) side view.

To reach the proposed design, it first passes through mainly two steps as shown in Figure 2a. First, a meander bowtie antenna is considered on a full ground plane (ANT I) which produces three frequency bands that require to be enhanced as depicted in Figure 2b. Then in (ANT II), a partial ground plane is utilized for better antenna characteristics that result in three operating bands in addition to other bands that required impedance matching. To optimize the matching impedance, the meandered arm's length, width, and spacing are considered as shown in (ANT III), which produces the proposed seven resonating bands with optimized rejection bands.

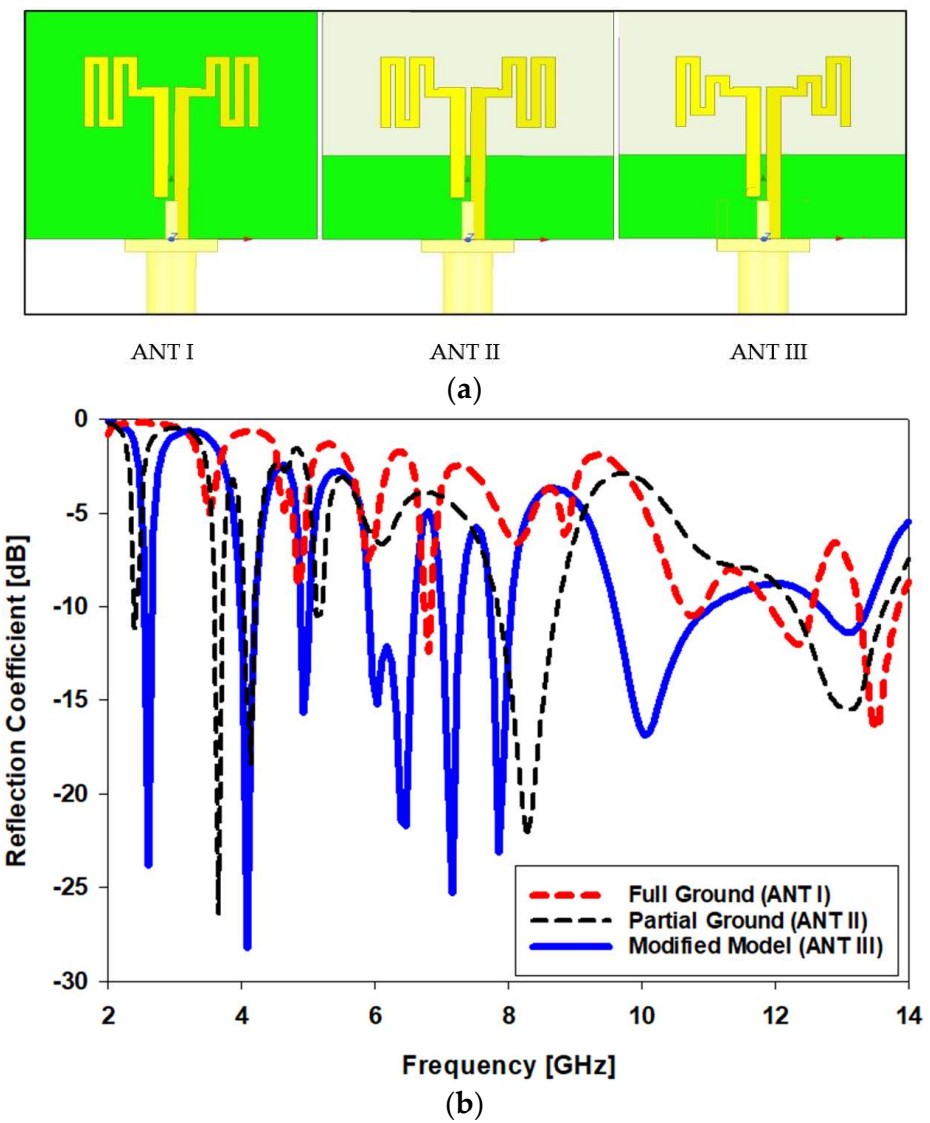

**Figure 2.** (**a**) Proposed antenna design steps; (**b**) reflection coefficient comparison for the design steps.

### 3. Results and Discussion

The designed antenna is built as shown in Figure 3a to validate the simulation results that are obtained through high-frequency structure simulator (HFSS) software, which were fine-tuned using an intense parametric optimization procedure to generate the reflection coefficient displayed in Figure 3b, where the simulated and measured $S_{11}$ are in good agreement. The antenna 6-dB bandwidth ranges from 2.52 GHz to 13.83 GHz, and its 10-dB bandwidth spans over 2.55–10.94 GHz, which satisfies UWB requirements. Due to the multi-band behavior of the proposed antenna; the 6-dB bandwidth can be considered to provide seven resonating bands. These bands have bandwidths of (2.52–2.68) GHz, (3.9–4.4) GHz, (4.9–5.2) GHz, (5.8–6.7) GHz, (6.8–7.4) GHz, (7.6–8.2) GHz, and (9.3–13.8) GHz, respectively, in addition to an acceptable 10-dB impedance bandwidth in both simulation and measurements if they were considered. Besides these resonating bands, the suggested antenna was also acquired with six frequency notches with a notch level $\geq -5$ dB. The rejecting bands' centered frequencies '$f_c$' are 3.24 GHz, 4.71 GHz, 5.48 GHz, 6.81 GHz, 7.51 GHz, and 8.7 GHz and the percentage bandwidths (PBW) are 41.36%, 14.65%, 15.15%, 5.29%, 5.99%, and 18.04%, respectively. Together, several resonances modes and notch-band techniques have been used in this design to pass or suppress different communication services.

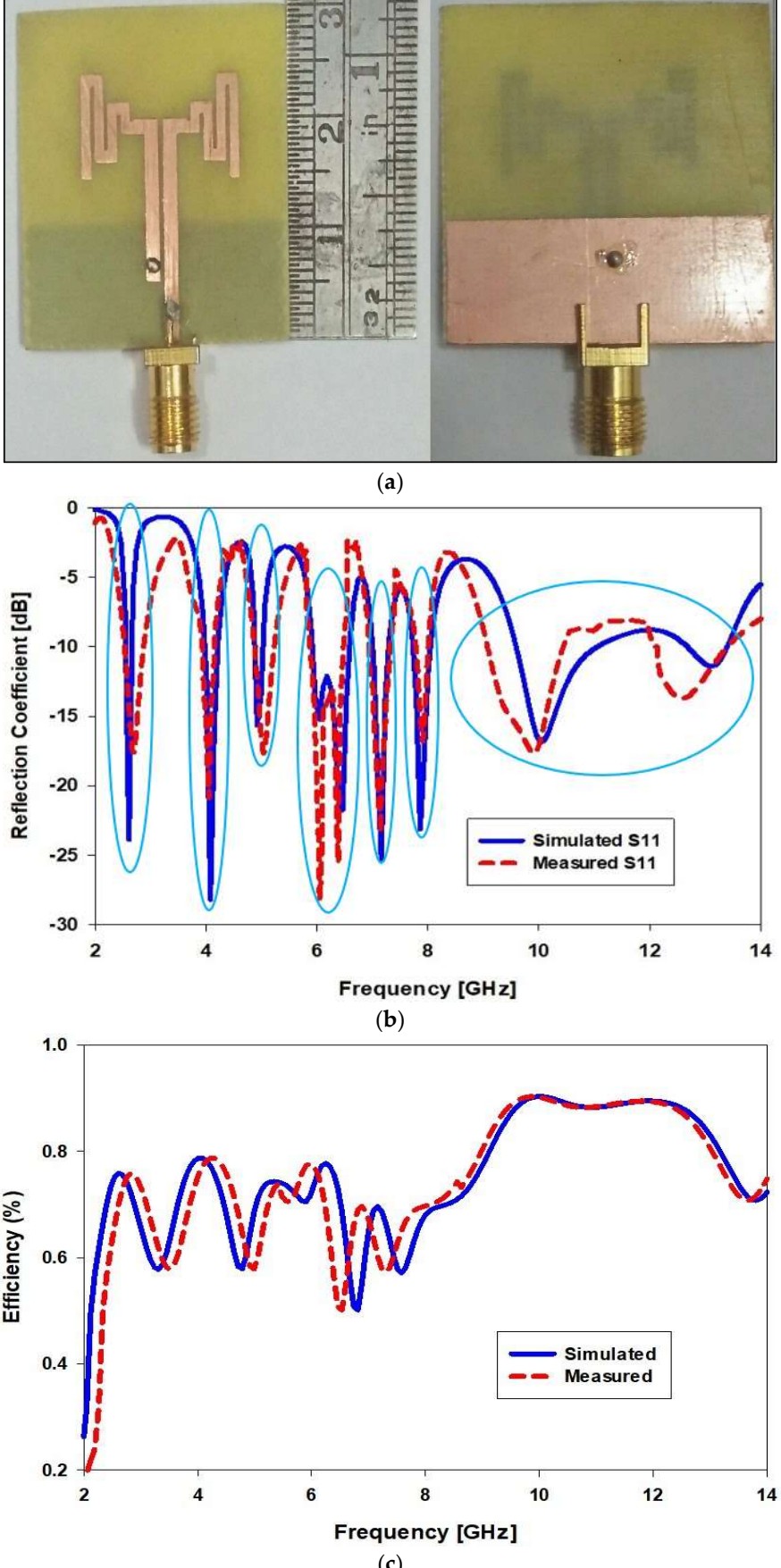

**Figure 3.** *Cont.*

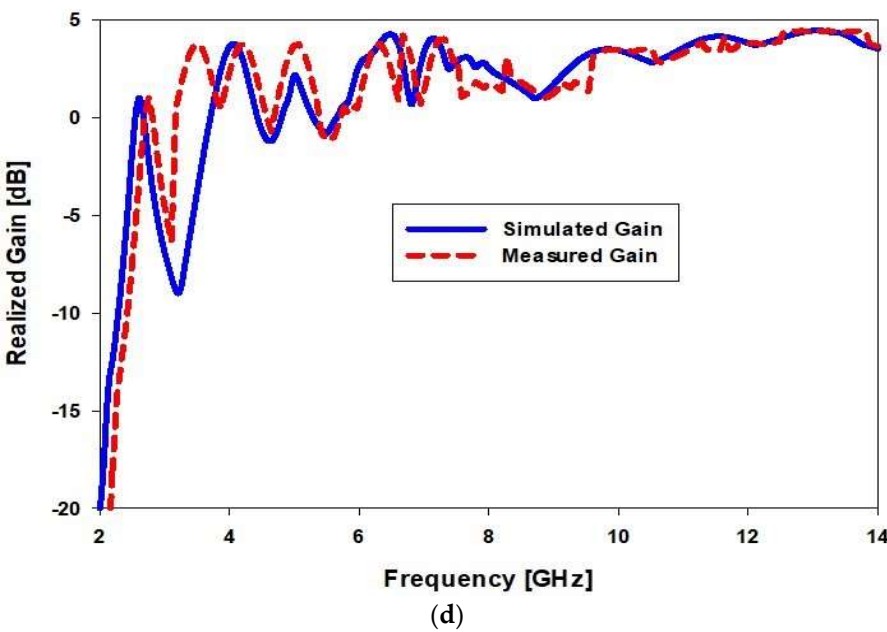

**(d)**

**Figure 3.** (**a**) The top and bottom layers of the fabricated proposed antenna, (**b**) simulated and measured antenna reflection coefficient, (**c**) proposed antenna design steps, and (**d**) reflection coefficient comparison for the design steps.

The simulated and measured peak realized gain graph is depicted in Figure 3c. A good agreement is noticed between the two traces where the gain tends to increase as the frequency increases with a sharp reduction at the rejecting band's center frequencies. The simulated results of the gain at the six notches' center frequencies are −8.2515 dB, −0.6843 dB, −0.2025 dB, 1.6513 dB, 2.5284 dB, and 0.78 dB, respectively. Table 2 summarizes the measured values of different aspects of the proposed antenna at these rejecting bands and the seven resonant frequencies. The proposed antenna peak realized gains within each resonance are tabulated in Table 2, where the maximum gain reaches 4.46 dB at 13.74 GHz. The graphs of the simulated and measured radiation efficiency are depicted in Figure 3d, and the results well agreed with the efficiencies attained at the operating band and a reduction at the rejecting bands. The maximum efficiency is found to be 90.3% at 9.82 GHz, while the lowest radiation efficiency reaches 50.14% at 6.53 GHz, which belongs to the fourth notch.

The fabricated antenna is tested in its characteristics by mounting the antenna in a suitable anechoic chamber to conduct the required measurements as shown in Figure 4a. The radiation pattern at all resonances-centered frequencies is measured and compared with the simulation. The pattern at eight selected frequencies in which the $S_{11}$ curve has deepest locations is plotted in Figure 4b–i for both the E-plane, which represent the YZ-plane ($\phi = 90°$), and the H-plane, which represents the XZ-plane ($\phi = 0°$). In the H-plane, the antenna has shown an omnidirectional pattern at the first two resonances and starts to exhibit directional radiation pattern towards single or multiple orientations. The E-plane demonstrates an omnidirectional radiation pattern at the first resonant while the beam starts to acquire different directional patterns at the other resonances, and this is due to the partial ground and other modes being involved as frequency was increased. A good agreement between the simulated and measured patterns are noticed for the shown patterns. The radiation pattern at the rejection band-centered frequencies is also monitored, and it has a similar behavior but with much less power compared to the resonating band, and the pattern at the first notch-centered frequency is plotted in Figure 4j.

**Table 2.** Measured characteristics of the proposed antenna at the notches.

| Notch No. | 10−dB BW (GHz) | $f_c$ | PBW % | Gain at $f_c$ |
|---|---|---|---|---|
| 1 | 2.88–3.93 | 3.1 | 33.87 | −6.35 |
| 2 | 4.15–4.88 | 4.64 | 15.73 | −0.69 |
| 3 | 5.23–5.88 | 5.55 | 11.71 | −1.20 |
| 4 | 6.44–6.97 | 6.95 | 7.63 | 0.70 |
| 5 | 7.29–7.82 | 7.58 | 6.99 | 1.07 |
| 6 | 8–9.08 | 8.84 | 12.22 | 0.98 |
| Measured Characteristics of the Proposed Antenna at the Resonances | | | | |
| Band No. | 6−dB BW (GHz) | $f_r$ | PBW % | Gain at $f_r$ |
| 1 | 2.43–3.03 | 2.7 | 22.22 | 1.02 |
| 2 | 3.71–4.23 | 4.05 | 12.84 | 3.52 |
| 3 | 4.76–5.38 | 5.05 | 12.28 | 3.77 |
| 4 | 5.83–6.54 | 6.04 | 11.75 | 3.82 |
| 5 | 6.85–7.44 | 7.15 | 8.25 | 4.06 |
| 6 | 7.56–8.01 | 7.90 | 5.78 | 3.5 |
| 7 | 9.27–13.88 | 11.55 | 39.83 | 4.46 |

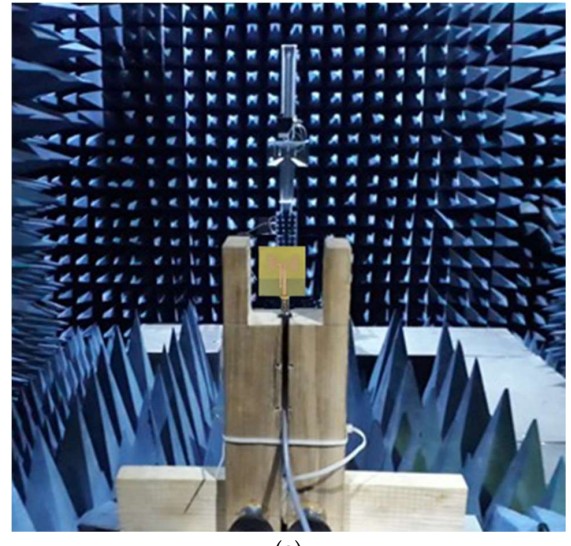

(**a**)

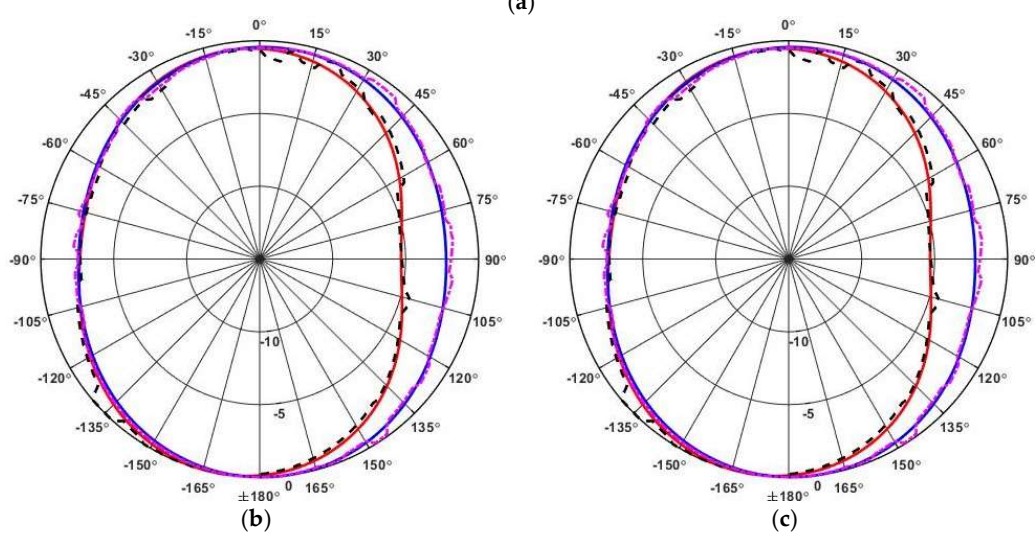

(**b**)　　　　　　　　(**c**)

**Figure 4.** *Cont*.

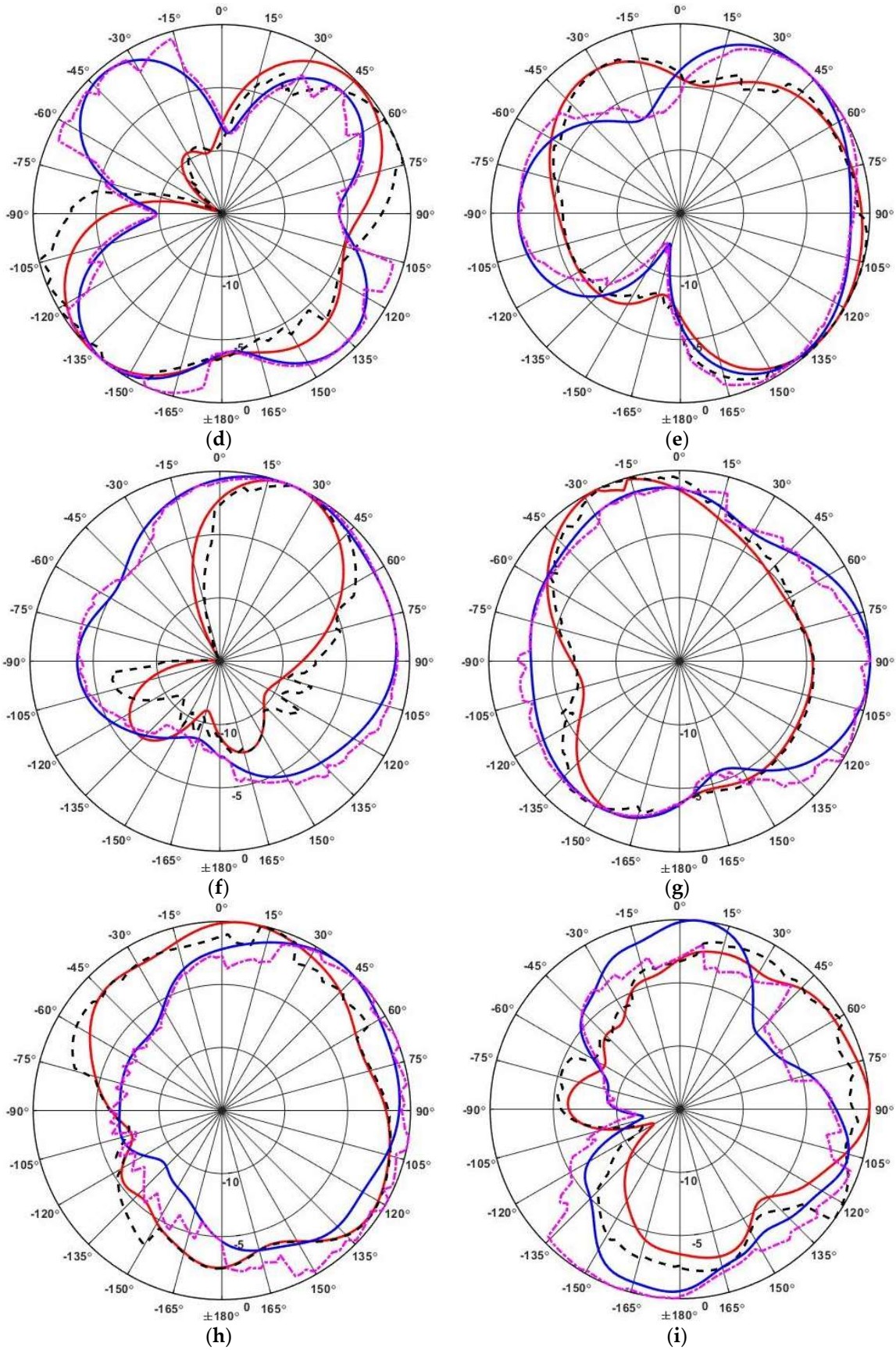

**Figure 4.** *Cont.*

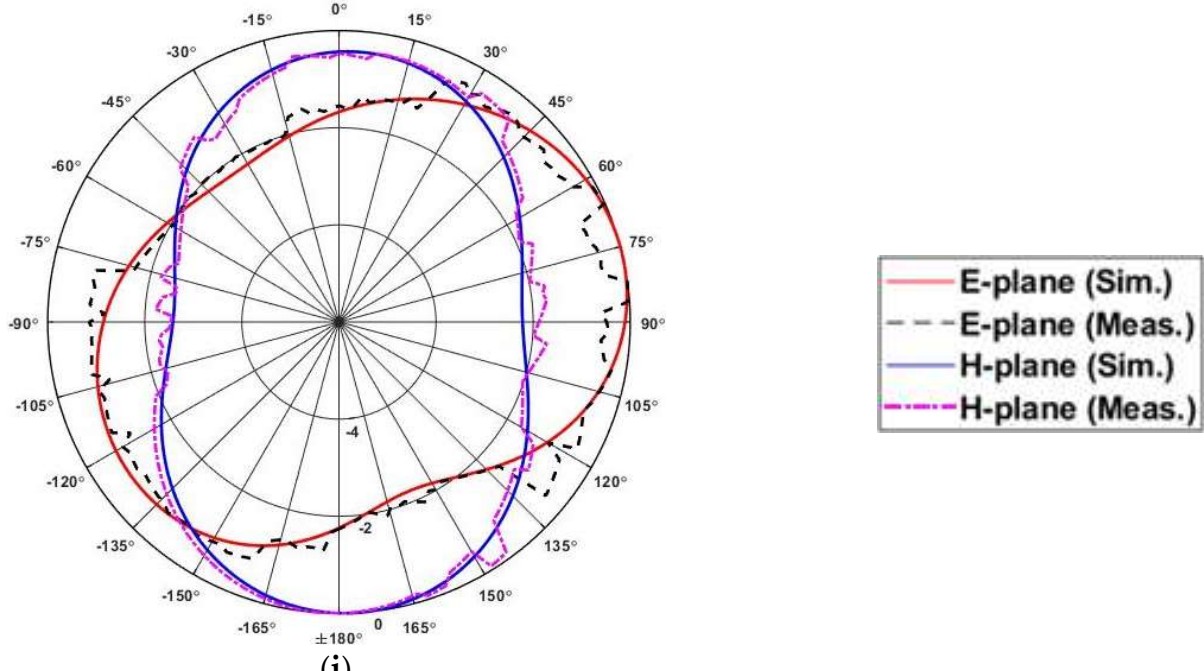

**(j)**

**Figure 4.** (**a**) Proposed antenna in the anechoic chamber. Simulated and measured antenna co-polarized radiation pattern for E- and H-planes at (**b**) 2.7 GHz, (**c**) 4.05 GHz, (**d**) 5.05 GHz, (**e**) 6.04 GHz, (**f**) 7.15 GHz, (**g**) 7.90 GHz, (**h**) 11.55 GHz, (**i**) 13.11 GHz, and (**j**) 3.1 GHz.

## 4. Time-Domain Characteristics

A significant aspect of UWB systems is the computation of the dispersion that happens when the antenna radiates and receives a pulse signal. The pulse-based UWB systems that utilize delivering very tiny pulses in time have a unique set of design requirements for antennas. As a result, more research and examination into the time domain behavior of a UWB antenna is needed. The pulse distortion and fidelity factor of the emitted pulse are the most important time-domain characteristics. They determine the amount of pulse distortion caused by the antenna.

In the proposed design, a normalized 5th order Gaussian derivative is set as an input signal. The pulse duration is 370 ps, as shown in Figure 5a, and it will have the wideband spectrum in the frequency domain as demonstrated in Figure 5b. The $n$th Gaussian pulse is represented in the time domain by Equation (2) where $H_n(t)$ is the $n$th Hermit polynomial, and its fifth-order polynomial is given in Equation (3) [25]:

$$G(t) = Ae^{-\frac{t^2}{2\sigma^2}} \tag{1}$$

$$G^n(t) = \frac{d^nG}{dt^n} = (-1)^n \frac{1}{\left(\sqrt{2}\sigma\right)^2} \cdot H_n\left(\frac{t}{\sqrt{2}\sigma}\right) \cdot G(t) \tag{2}$$

$$H_5(t) = 32t^5 - 160t^3 + 120t \tag{3}$$

The Fourier transform of the input signal is depicted in Figure 5c and resembles the reflection coefficient generated in the frequency domain and described in Figure 3b. The output normalized signal is shown in Figure 5d, and it has a pulse width of 3.74 ns, in which the signal distortion is around 50% in the first 1 ps; then, it reduces rapidly after that.

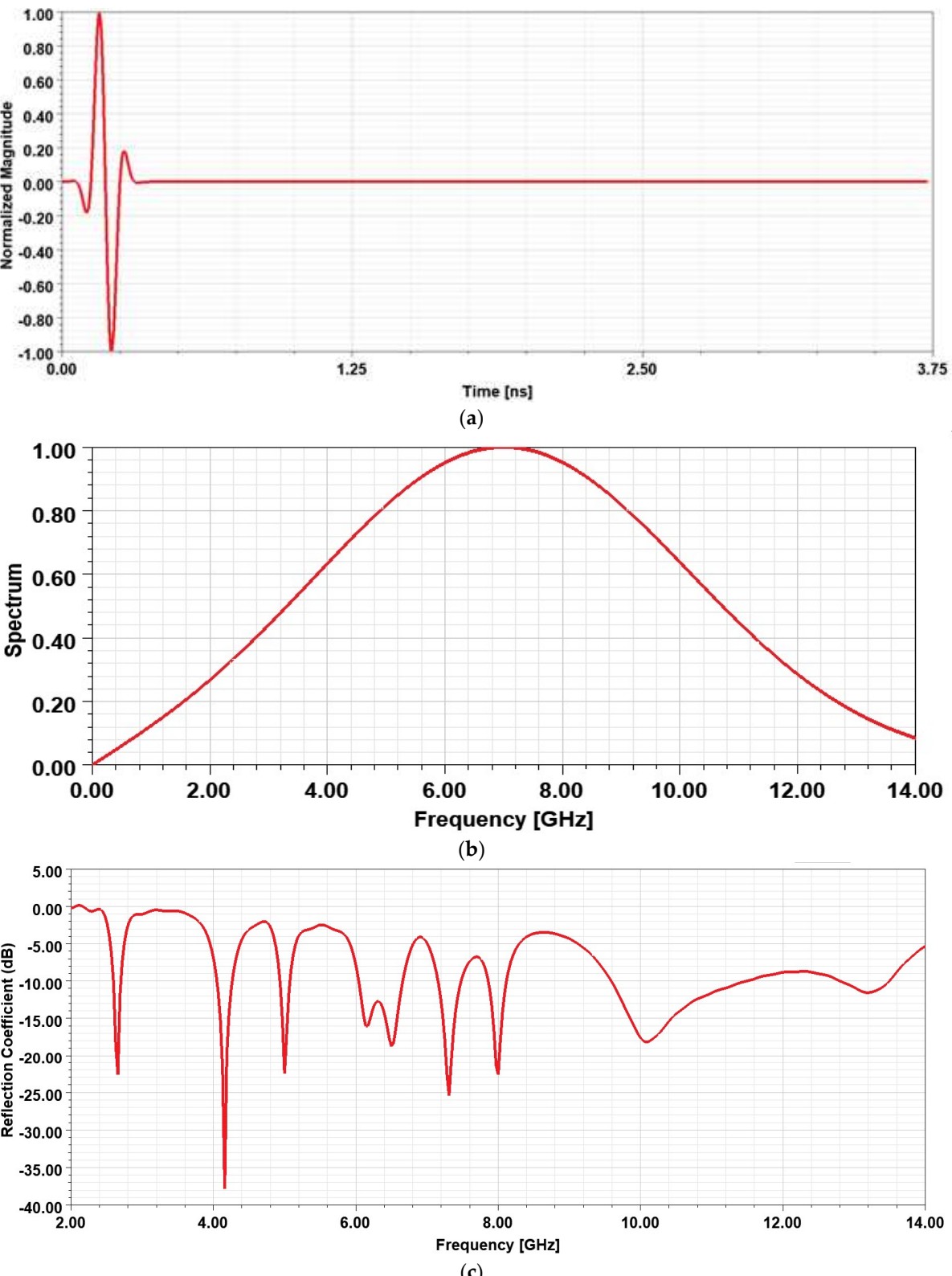

**Figure 5.** *Cont.*

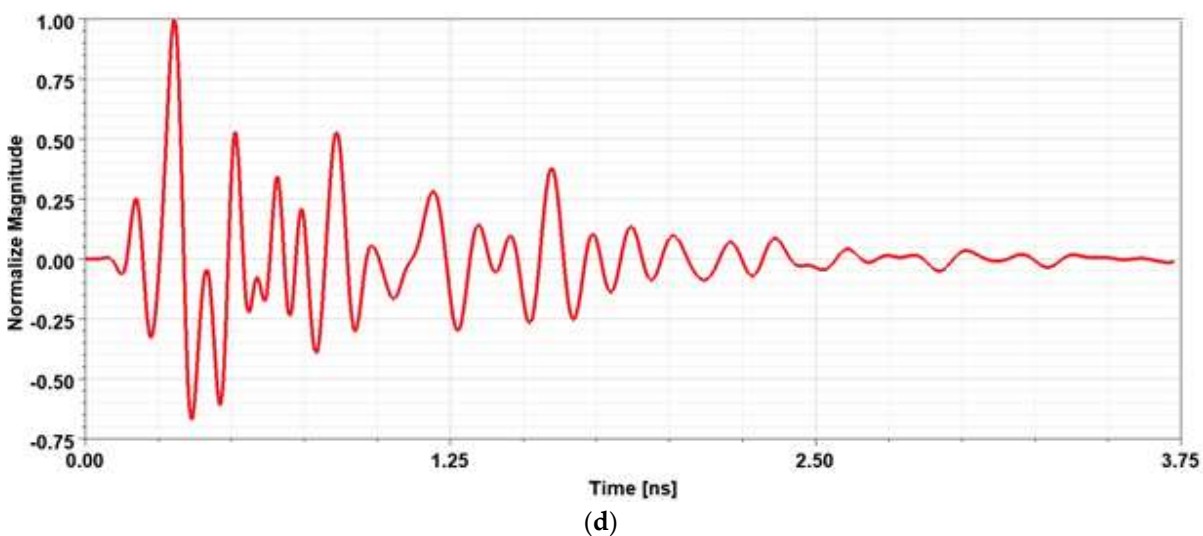

**(d)**

**Figure 5.** Time-domain results of the proposed antenna showing (**a**) input pulse, (**b**) input spectrum, (**c**) Fourier transform of the input signal, and (**d**) output signals.

## 5. Equivalent Circuit Analysis

Creating an equivalent circuit of lumped components for such a design analyzed in this paper is not an easy process. First, the antenna real and imaginary input impedance must be monitored to predict the best resonator to be used and whether they have common behavior or not. As can be noticed from Figure 6, the first six resonances have an impedance close to 50 Ω near the resonating frequency, while for the last wideband, the impedance is widely spanned near the 50 Ω line. An assumed circuit model is displayed in Figure 7, where the inductance (Lf) represents the input feedline and the first parallel RLC model to match the shorting pin used in the design; then, several series resonators are employed to create the first six bands, which are representing the meandered bowtie arms. Finally, the partial ground plane, which has a major effect in the wideband of the seventh band, is modelled by the rest of the circuit.

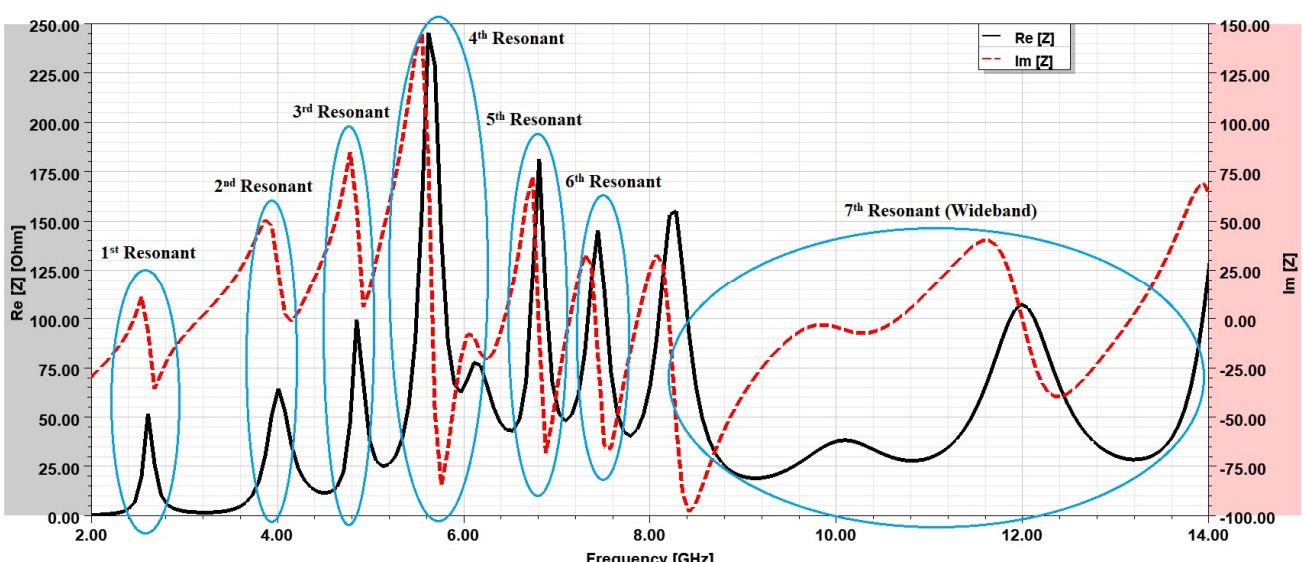

**Figure 6.** Real and imaginary impedance for the designed model.

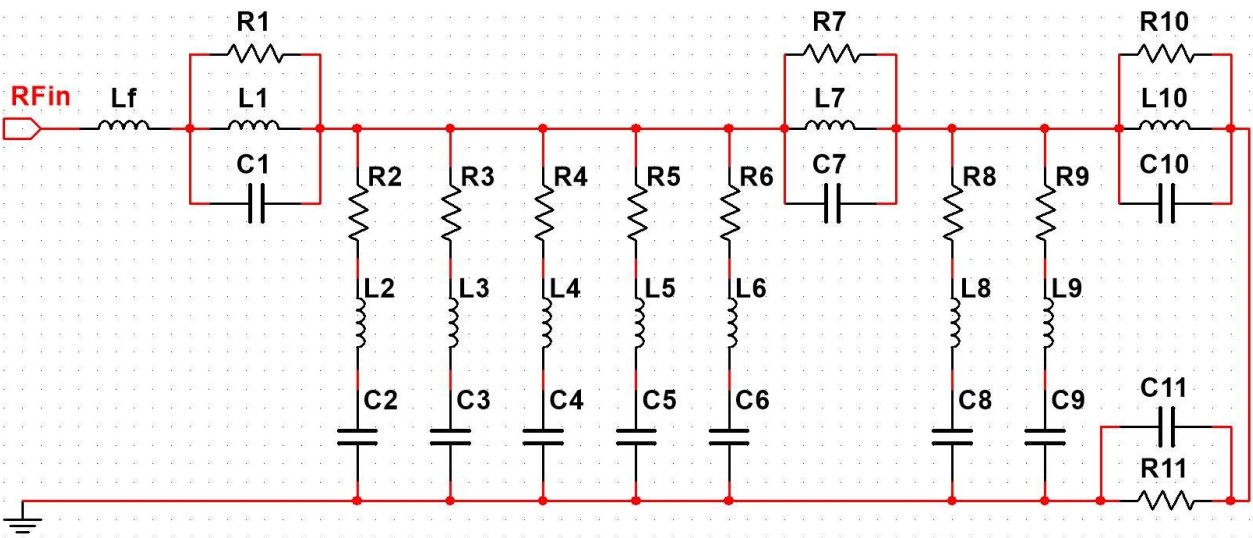

**Figure 7.** Equivalent circuit model.

The proposed circuit model is optimized using an ADS environment and has an approximated S-parameter result matched favorably with the simulated and measured results discussed before, and it is shown in Figure 8. The optimized values of the lumped circuit components are summarized in Table 3.

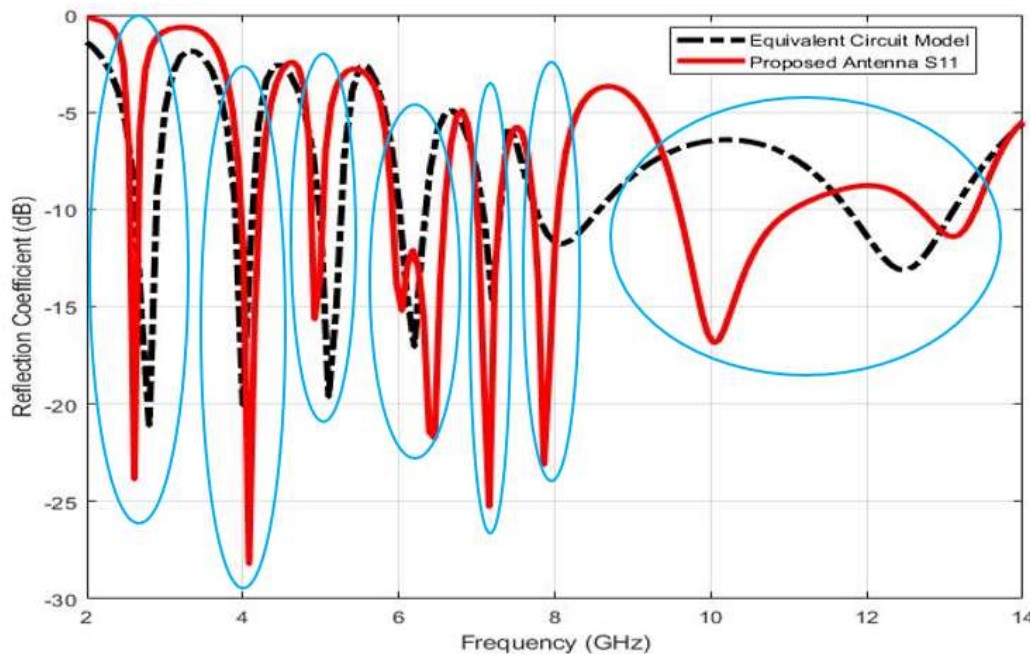

**Figure 8.** Comparison between simulated and calculated reflection coefficient.

**Table 3.** Comparison between the simulated and calculated Q and *ROC* of the proposed antenna.

|  | Notch 1 | Notch 2 | Notch 3 | Notch 4 | Notch 5 | Notch 6 |
|---|---|---|---|---|---|---|
| *Q* Simulated | 3 | 9.47 | 7.51 | 21.28 | 19.76 | 7.37 |
| *Q* Calculated | 4.53 | 6.92 | 9.33 | 11.17 | 18.5 | 3.55 |
| *ROC* Simulated | 7.68 | 11.32 | 8.09 | 12.43 | 9.71 | 4.11 |
| *ROC* Calculated | 8.1 | 8.23 | 10.45 | 7.25 | 8.57 | 1.12 |

Two elements, the quality factor ($Q$) that represents the ratio of the stored to the dissipated energy in each cycle, and the roll-off criteria ($ROC$) which represent the variation in the reflected signal versus frequency, are completely linked to the selectivity and sharpness of the rejection mechanism. For the frequency rejection bands, the quality factor of the RLC series and parallel resonators can be calculated as [26].

$$Q = \frac{f_c}{BW_{-3dB}} \tag{4}$$

$$ROC = \frac{\Delta S_{11}(dB)}{\Delta f(GHz)} \tag{5}$$

where $BW_{-3dB}$ represents the notch bandwidth at a level of 3 dB below its maximum, and $\Delta S_{11}$(dB) is the variation in the reflection coefficient within the frequency band represented by $\Delta f$(GHz). The calculated quality factor and $ROC$ of the six rejection bands based on Equations (4) and (5) are summarized in Table 4.

**Table 4.** Lumped component values of the equivalent circuit model (R in $\Omega$, C in pF and L in nH).

| R1 = 28.81 | R2 = 37.43 | R3 = 55.52 | R4 = 120.53 | R5 = 27.31 | R6 = 114.96 |
|---|---|---|---|---|---|
| L1 = 3.80 | L2 = 31.96 | L3 = 47.75 | L4 = 5.52 | L5 = 40.02 | L6 = 147.98 |
| C1 = 0.14 | C2 = 0.095 | C3 = 0.031 | C4 = 0.052 | C5 = 0.022 | C6 = 0.033 |
| R7 = 3.38 k | R8 = 70.24 | R9 = 81.69 k | R10 = 9.21 k | Lf = 1.84 | |
| L7 = 902.61 | L8 = 34.62 | L9 = 26.96 | L10 = 1.09 m | R11 = 0.35 | |
| C7 = 1.98 | C8 = 0.018 | C9 = 1.98 μ | C10 = 0.18 | C11 = 0.58 | |

## 6. Quad Port MIMO Antenna Analysis

The multiband single element antenna proposed is expanded to a four-port MIMO antenna place such that the adjacent elements are orthogonal to each other. The substrate size is $70 \times 70$ mm$^2$ and 10 mm distance between orthogonal antenna elements, as shown in Figure 9. The current distribution at the center frequency of the six rejection bands is plotted and shown in Figure 10a–f to determine the antenna portion that is responsible for each notch. The first notch, at a center frequency of 3.24 GHz, is caused by the first two arms of the right bowtie portion, whereas the left feed line and the left bowtie portion have a dominant effect for creating the second notch at 4.64 GHz. At 5.48 GHz, which is the center of the third notch, the right feed line and the last arm of the bowtie are dominant. For the last three notches at 6.81 GHz, 7.51 GHz, and 8.7 GHz, the responsible potions are the lower portion of the feed line, the two feedlines, and the middle portion of the feedlines, respectively.

The total active reflection coefficient (TARC) of the four-port MIMO antenna is depicted in Figure 11. TARC relates the total incident power to the total radiated power, and it is noticed that the MIMO antenna has a TARC similar to the reflection coefficient of the single element with a similar number of resonating bands and center frequencies. The mutual coupling between all four ports is plotted in Figure 12. The average mutual coupling between the elements at the operating bands center frequency is below −17 dB across the seven resonating bands, while it is a little higher (−13 dB) at the first resonant between the adjacent elements, and it is much lower between diagonal elements.

The Envelope Correlation Coefficient (*ECC*) and diversity gain (*DG*) are useful metrics for assessing the diversity of MIMO antenna elements [4]. The ECC (much lower than 0.5) and DG ($\cong$10 dB) of the proposed MIMO are displayed in Figure 13, and they are calculated from the scattering parameters between the two ports of high mutual coupling as given by Equations (6) and (7), respectively. Other MIMO characteristics such as mean effective gain (MEG) and channel capacity loss (CCL) for the seven resonating bands are tabulated in Table 5, and the results are satisfying the determined limit for these quantities, such as

the CCL, is below 0.4 bit/s/Hz; the MEG ratios between the investigated ports are close or equal to unity.

$$ECC = \frac{|S_{11} * S_{12} + S_{21} * S_{22}|^2}{\left(1 - |S_{11}|^2 - |S_{21}|^2\right)\left(1 - |S_{22}|^2 - |S_{12}|^2\right)} \tag{6}$$

$$DG = 10\sqrt{1 - |ECC|^2} \tag{7}$$

**Table 5.** Proposed MIMO antenna characteristics at the resonant frequencies.

|  | MEG 1 (dB) | MEG 2 (dB) | MEG 1/MEG 2 | CCL (bit/s/Hz) |
|---|---|---|---|---|
| Resonant 1 | −3.57 | −3.52 | 0.99 | 0.38 |
| Resonant 2 | −3.09 | −3.12 | 1.01 | 0.06 |
| Resonant 3 | −3.22 | −3.21 | 1 | 0.14 |
| Resonant 4 | −3.27 | −3.27 | 1 | 0.18 |
| Resonant 5 | −3.07 | −3.03 | 0.99 | 0.03 |
| Resonant 6 | −3.05 | −3.05 | 1 | 0.03 |
| Resonant 7 | −3.15 | −3.16 | 1 | 0.09 |

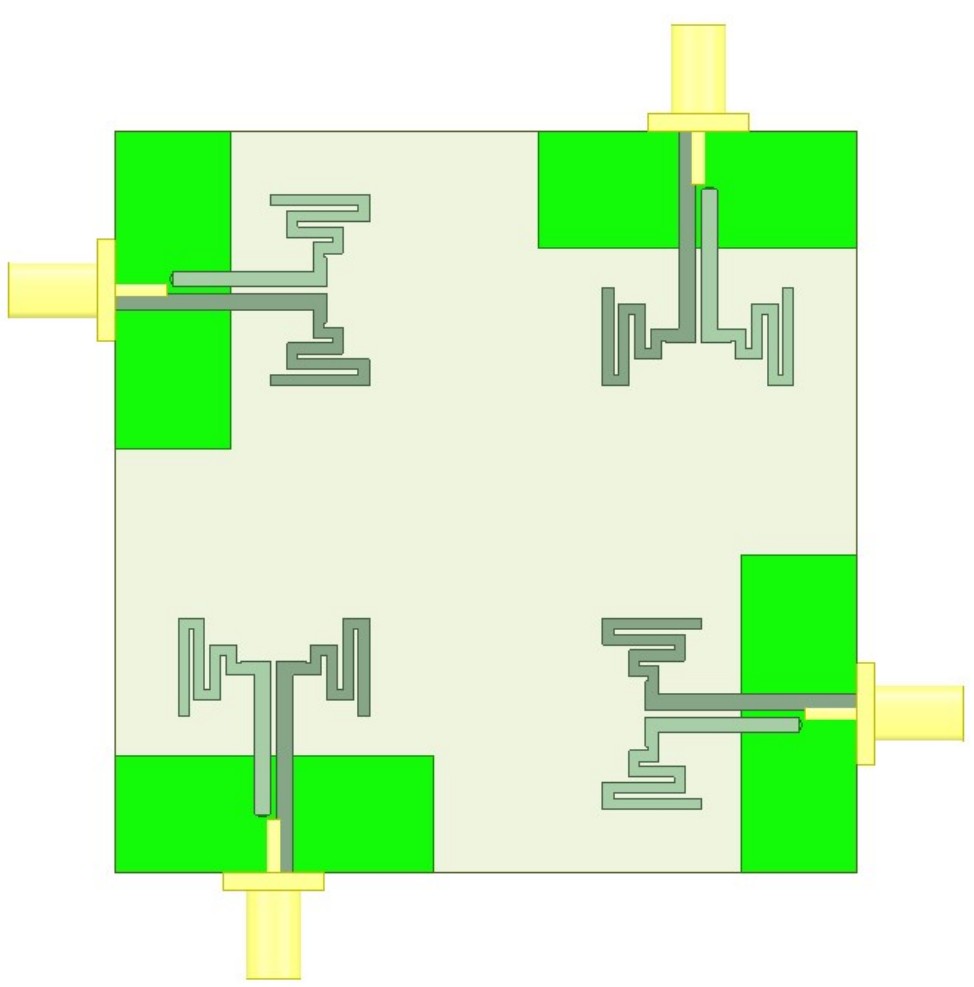

**Figure 9.** Four-element MIMO antenna configuration.

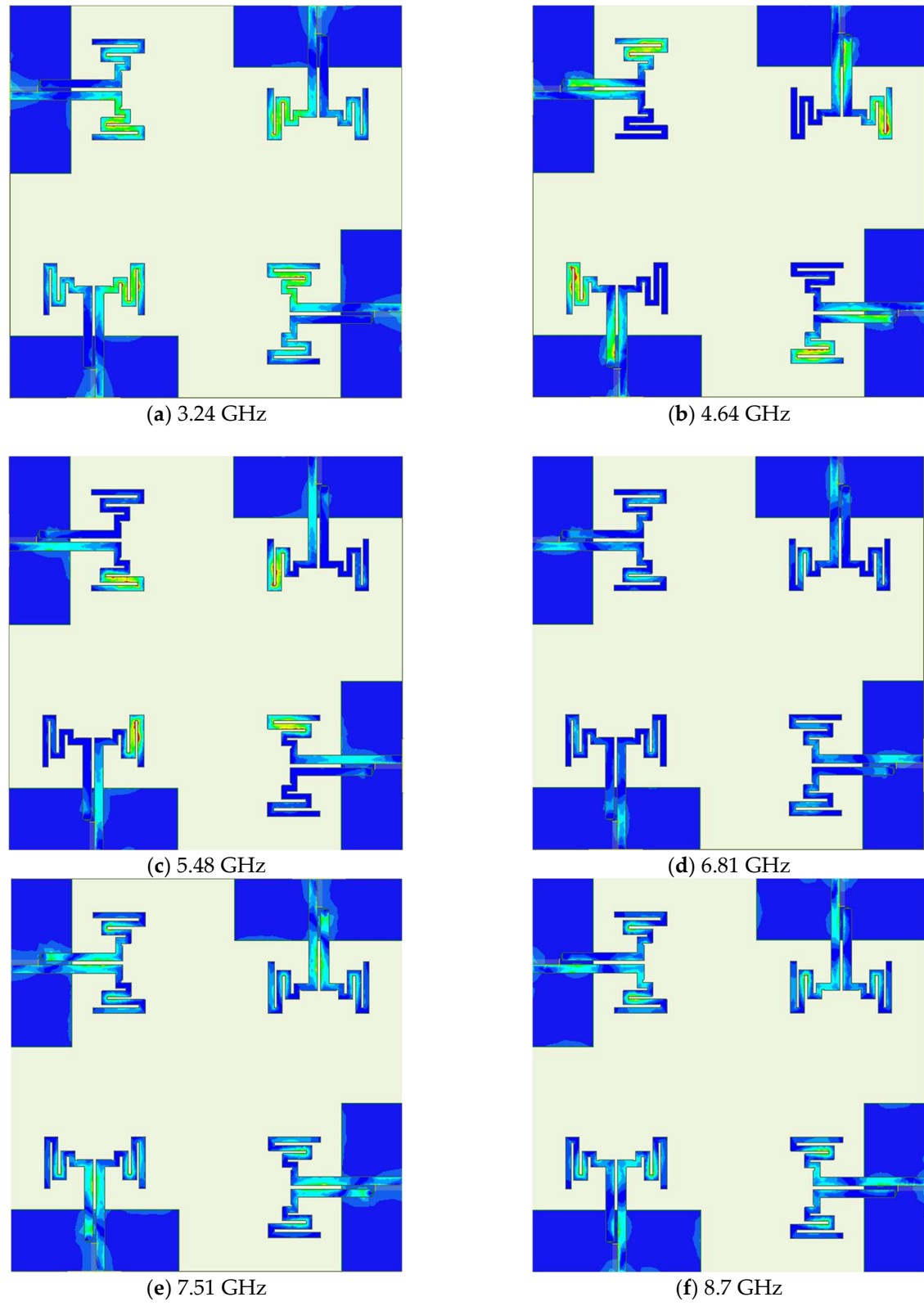

**Figure 10.** Current distribution at the rejection bands center frequency; (**a**) at 3.24 GHz, (**b**) at 4.64 GHz, (**c**) at 5.48 GHz, (**d**) at 6.81 GHz, (**e**) at 7.51 GHz, (**f**) at 8.7 GHz.

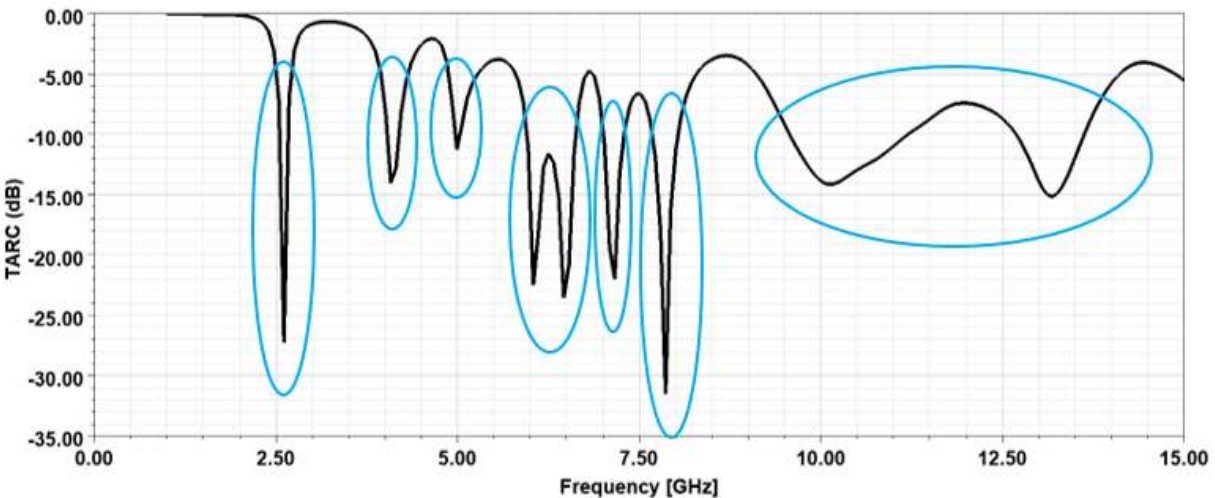

**Figure 11.** The total active reflection coefficients of the MIMO antenna.

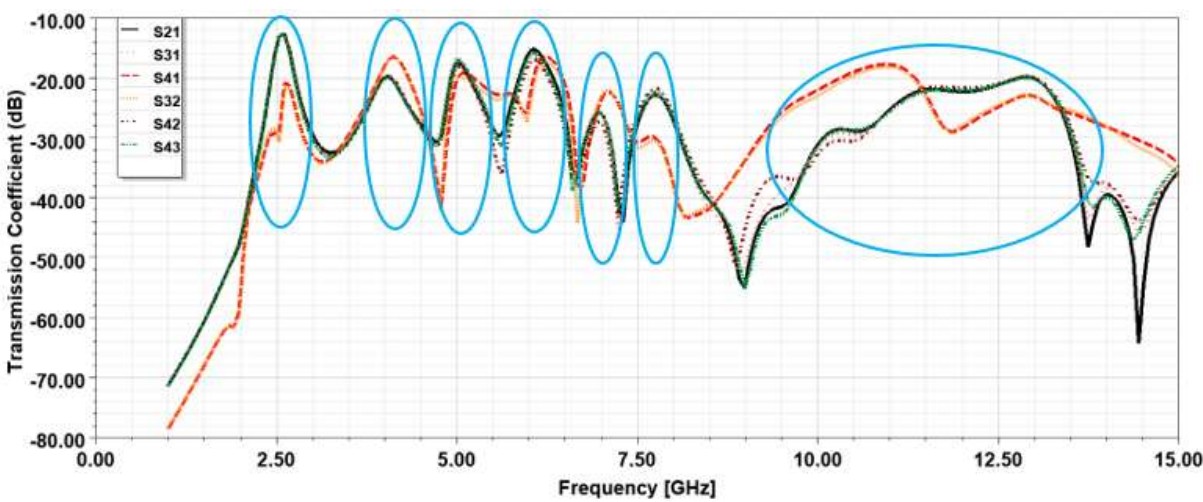

**Figure 12.** The transmission coefficients between all four ports.

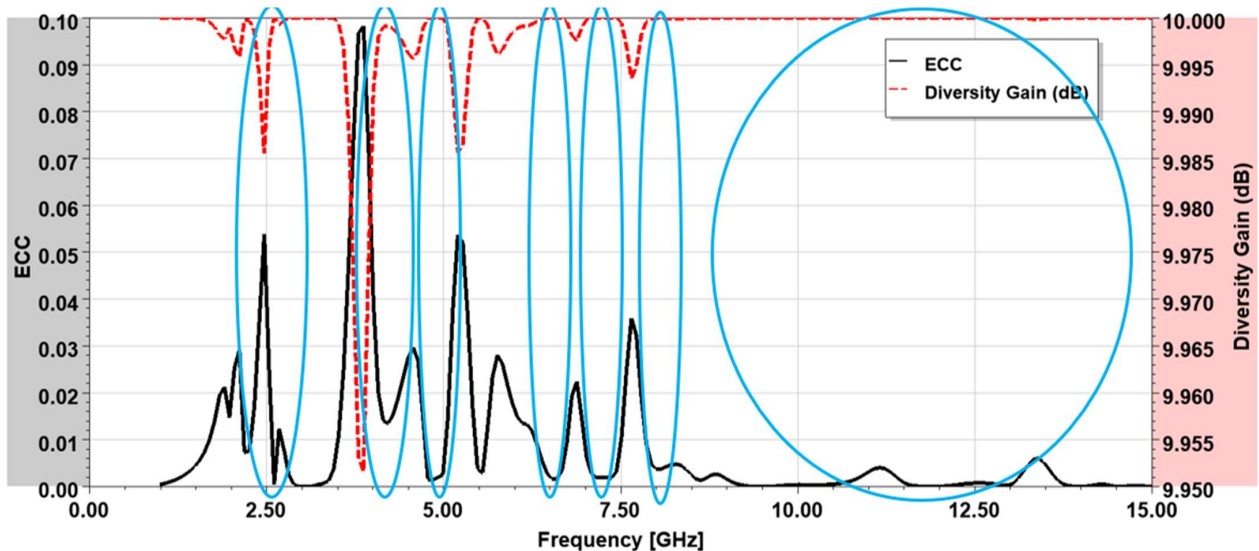

**Figure 13.** ECC and DG for the four-port MIMO Design.

## 7. Comparison with Related Works and Served Applications

The proposed antenna results are compared with the related works available in the literature for different aspects and the comparisons are tabulated in Table 6. The proposed antenna has a relatively smaller size and can pass low frequencies due to the meander arm structure. Additionally, the proposed antenna can operate at seven distinguished bands by satisfying the UWB license range. The maximum gain is acceptable for such a monopole antenna, and it is lower than some works in the literature due to the multiband mode of operations.

**Table 6.** Comparison of this work with other multiband antennas.

| Ref. | Radiator Shape | Resonances (GHz) | Gain (dB) | Size (mm²) | No. of Spectrums | Covered Bands |
|---|---|---|---|---|---|---|
| [16] | Split Ring | 2.45, 5.2, 9.7 | 7 | 40 × 30 | 3 | S, C, X |
| [17] | Split Ring metamaterial | 4.27, 5.42, 12.4 | - | 20 × 20 | 3 | C, X |
| [18] | Bowtie and meandered lines | 1.62, 4.22, 7.13 | 8.2 | 20 × 10.4 | 3 | L, C |
| [19] | Triangle and V-parasitic | 2.88, 3.64, 3.95, 4.38, 4.81, 5.6 | 10.5 | 70 × 50 | 6 | S, C |
| [20] | Slotted conical patch | 2.4, 5.2, 5.8, 27.5 | 5.85 | 30 × 30 | 4 | S, C, Ka |
| [21] | F-shape element on a truncated ground plane | 2.1, 2.4, 3.35, 3.5, 5.28, 5.97 | 3.88 | 40 × 35 | 6 | S, C |
| Proposed Antenna | Meandered Bowtie | 2.7, 4.05, 5.05, 6.04, 7.15, 7.9, 11.55 | 4.46 | 30 × 30 | 7 | S, C, X |

## 8. Conclusions

The proposed design can be used for broadband radio applications within the first resonance, long-distance radio telecommunications; sub-6 GHz and other C-Band applications can be served by utilizing the second to fifth spectrums; X-Band radar applications can be supported by the sixth spectrum (7.6–8.2) GHz, and space communications, terrestrial broadband, satellite communications, amateur radio systems aided by the last vast spectrum (9.27–13.83) GHz. In addition, it can support the fifth generation (5G) FR1 band that is intended to cover up to 7.125 GHz. The achieved notches are intended to avoid interfering with WiMAX through the first notch, aeronautical radio navigation, which falls within the second and the sixth notches, the 5 GHz Wi-Fi band within the third rejecting band, and fixed and mobile satellite transmission within the fourth and fifth stopbands.

A printed microstrip meandered bowtie antenna is investigated and fabricated on an FR-4 substrate. The upper layer contains the two portions of the bowtie, where the right segment of the bowtie is coupled to the feedline, and the left portion is linked to the ground in the back layer through a shorting pin. Although the structure seems small and simple, the antenna exhibits seven resonating bands and six rejecting bands by optimizing the meandered arm's length, width, number, and spacing, making the antenna suitable for sub-6 GHz and UWB 5G communications. The antenna has a maximum gain and radiation efficiency of 4.46 dB and 90.3%, respectively. The MIMO configuration of the proposed antenna shows an acceptable value for ECC, DG, MEG, and CCL. The antenna is best suited for an end-user customer to accept and reject different communication services by just utilizing the corresponding band.

**Author Contributions:** Conceptualization, Y.S.F. and S.A.; methodology, Y.S.F. and S.A.; software, Y.S.F.; validation, Y.S.F., N.O.P., C.H.S., R.A.-A. and S.A.; formal analysis, N.O.P., C.H.S., R.A.-A. and Y.S.F.; investigation, Y.S.F. and S.A.; resources, Y.S.F. and S.A.; data curation, Y.S.F. and R.A.-A. writing—original draft preparation, Y.S.F.; writing—review and editing, Y.S.F., C.H.S. and S.A.; visualization, Y.S.F., N.O.P., C.H.S. and R.A.-A.; supervision, Y.S.F.; project administration, Y.S.F.; funding; N.O.P., C.H.S., R.A.-A. All authors have read and agreed to the published version of the manuscript.

**Funding:** This paper is also partially funded by British Council "2019 UK-China-BRI Countries Partnership Initiative" program, with project titled "Adapting to Industry 4.0 oriented International Education and Research Collaboration.

**Data Availability Statement:** The data is available with the author upon request.

**Acknowledgments:** The authors would like to thank the Deanship of Scientific Research at the University of Jordan for providing the facilities for conducting this research for the year 2019–2022. The authors wish to express their thanks to the support provided by British Council "2019 UK-China-BRI Countries Partnership Initiative" programme with project titled "Adapting to Industry 4.0 oriented International Education and Research Collaboration".

**Conflicts of Interest:** The authors declare no conflict of interest.

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
