# Peer review of "A Novel Meander Bowtie-Shaped Antenna with Multi-Resonant and Rejection Bands for Modern 5G Communications"

_electronics, doi:10.3390/electronics11050821_

Round 1

Reviewer 1 Report

The authors presented a Novel Meander Bowtie-Shaped Antenna with Multi-Resonant and Rejection Bands for Modern 5G Communications.

I appreciate the work that the authors have been done. However, this article missed some of the essential principles that need to be solved, as follows:

1- The authors are required to summarize the literature with the recently published articles for UWB planar antennas with band notch rejections. Please add more references about the planar antennas and explain some of their performance in terms of (size, gain, etc.) [1-3]

[1] UWB Dual-Band-Notched Lanky-Leaf-Shaped Antenna with Loaded Half-Square-Like Slots for Communication System" Electronics 10, no. 16: 1991. 

[2] A Planar Four-Element UWB Antenna Array with Stripline Feeding Network" Electronics 11, no. 3: 469. 

[3] High Gain of UWB Planar Antenna Utilising FSS Reflector for UWB Applications, cmc-computers, materials & continua, 2022, 70(1), 1419–1436.

2- I suggest adding at least one radiation pattern at the Rejection Bands. 

3- Talk about MIMO results in the conclusion. 

4- Add the abbreviation 'TARC' to line 254.

5- Modify (CCL0 to (CCL) in line 373.

6- Modify citation on the first page to include all the authors.

7- Add a reference for equations 5 and 6.

Author Response

Answers to the Reviewer#01:

We are thankful to the reviewer for mentioing these comments. Here we have addressed all the comments and the answers are given below:

  1. The suggested three papers have been addressed to enhance the introduction section with more recent and related works.
  2. The radiation pattern at the first notch is added in Fig. 4(j)
  3. MIMO findings have been summarized in the conclusion.
  4. The full name of TARC has been added.
  5. The typo mistake has been corrected as suggested.
  6. The citation on the first page has been corrected.
  7. A reference is added for the mentioned equations.

Reviewer 2 Report

In this paper a bowtie-shaped antenna with multi operating bands is designed, fabricated and measured. The proposed antenna has a compact size, simple structure, acceptable gain and seven operating frequencies. This is nice design but there are some comments:

1-The bowtie-shaped antenna previously published in [R1]. In the previous published conference, only simulated bowtie-shaped antenna was proposed and in this manuscript, fabricated device is proposed. Only one author is same in two papers. Many parts of manuscript are copied from [R1], should be revised. Especially abstract part should be rewritten. The plagiarism check file is attached.

2-Novelty of the proposed paper should be emphasized in the text and improved parts compared to the [R1] should be listed briefly to reviewers and editor.

3-Introduction section should be improved. some resonator with meandered structures can be briefly, discussed in the introduction like:” A Wilkinson power divider with harmonics suppression and size reduction using meandered compact microstrip resonating cells. Frequenz, 2017” and “A compact coupler design using meandered line compact microstrip resonant cell (MLCMRC) and bended lines. Wireless Networks, 2021”.

4- Seven operating bands should be emphasized with shaded region in Figures of 3, 8, 11, 12 and 13.

5- Figure 2, not cited in the text. Sufficient explanation about this figure should be added.

6- Figure 7, not cited in the text. Sufficient explanation about this figure should be added.

7- The mutual coupling effect in Four element MIMO antenna configuration should be investigated.

[R1] Ali Salim, Saleh Baqaleb, Yanal S. Faouri. "Multiband Meander UWB Bowtie Antenna with Six Rejection Bands", 2020 11th International Conference on Information and Communication Systems (ICICS), 2020.

Author Response

Answers to the reviewer#02:

  1. We appreciate the reviewer's help in providing the similarity report. The revised manuscript has been rewritten and the similarity is now 15%.
  2. The submitted manuscript has extremely expanded compared to the mentioned [R1] by including design steps for the proposed antenna, fabricated prototype and measurements results, Time-domain analysis, Equivalent circuit model and circuit parametric effects, four-element MIMO configuration and MIMO analysis such as mutual coupling, ECC, DG, MEG, CCL, TARC, current distribution at all notched bands center frequencies and comparison with recent related published works.

z The introduction has been enhanced by mentioning other related usages of meander line structure and the suggested two papers were helpful.

  1. The operating bands in the mentioned figures have been highlighted.
  2. Figure 2 has been explained.
  3. Figure 7 has been further explained.
  4. The mutual coupling has been further explained and plotted in Fig. 12.

Round 2

Reviewer 2 Report

The authors have addressed all my comments.